# A Critical Evaluation of AI Feedback for Aligning Large Language Models

**Archit Sharma**[*] **Sedrick Keh**[†] **Eric Mitchell**[*]

**Chelsea Finn**[*] **Kushal Arora**[†] **Thomas Kollar**[†]

[*]Stanford University [†]Toyota Research Institute
architsh@cs.stanford.edu

## Abstract

Learning from AI feedback (LAIF) is a popular paradigm for improving the instruction-following abilities of powerful pre-trained language models. LAIF first performs supervised fine-tuning (SFT) using demonstrations from a teacher model and then further fine-tunes the model with reinforcement learning (RL) or direct preference optimization (DPO), using feedback from a critic model. While recent popular open-source models have demonstrated substantial improvements in performance from the RL step, in this paper we question whether the complexity of this RL step is truly warranted for AI feedback. We show that the improvements of the RL step are virtually entirely due to the widespread practice of using a weaker teacher model (e.g. GPT-3.5) for SFT data collection than the critic (e.g., GPT-4) used for AI feedback generation. Specifically, we show that simple supervised fine-tuning with GPT-4 as the teacher outperforms existing LAIF pipelines. More generally, we find that the gains from LAIF vary substantially across base model families, test-time evaluation protocols, and critic models. Finally, we provide a mechanistic explanation for when SFT may outperform the full two-step LAIF pipeline as well as suggestions for making LAIF maximally useful in practice. Code is available at: https://github.com/architsharma97/dpo-rlaif.

## 1 Introduction

As the raw capabilities of open-source large language models (LLM) improve through pre-training at a large scale [Touvron et al., 2023a,b, Jiang et al., 2023, 2024, Bai et al., 2023, Bi et al., 2024], methods for effectively 'aligning' these models; i.e., steering them to follow user instructions effectively and safely, has garnered increasing attention. Supervised fine-tuning (SFT) with large datasets of user queries and human-written responses is one popular approach. Further refining a model fine-tuned with SFT using reinforcement learning from human feedback (RLHF; Christiano et al. [2017]) has been shown to further improve the quality of model responses, as judged by humans [Ouyang et al., 2022, Stiennon et al., 2020]. However, collecting the data for SFT and RLHF is expensive, requiring human annotations for both the SFT and preference rankings over several candidate responses to a query to be used as feedback for the RL stage. Given the high cost of data collection as well as the high level of disagreement among human annotators, many works such as Bai et al. [2022] and Lee et al. [2023] have replaced the human annotators in both the SFT and RLHF stages with strong *language model* annotators such as GPT-4 [Achiam et al., 2023]. The corresponding techniques, supervised model distillation [Taori et al., 2023, Chiang et al., 2023, Ding et al., 2023] and learning from *AI* feedback (LAIF; Bai et al. [2022], Lee et al. [2023], Cui et al. [2023]), have proven effective in training state-of-the-art open source models [Tunstall et al., 2023, Ivison et al., 2023, Intel, 2023,

Teknium, 2023]. However, plugging in LLMs to substitute human annotators in the same RLHF pipeline may not be the best way to leverage such LLMs. First, LLMs are often better at *generating answers* for instructions than discriminative tasks [West et al., 2023], and labeling preferred answers for AI feedback is an example. Further, while humans may find comparing answers incurs lower cognitive cost than producing answers themselves, for modern LLMs, the cost of generating a preference comparison or ranking may actually be higher than simply generating a demonstration, owing to the longer input context when comparing multiple responses. Therefore, while LAIF may seem like a more convenient variant of RLHF prima facie, we ask: *what kind of data from strong LLMs is more effective for learning instruction-following: completions for SFT or AI feedback for RL?*

To answer this question, we compare the effectiveness of the LAIF pipeline with doing SFT on demonstrations directly generated from the annotator language model. In our experiments, we align a variety of pre-trained base language models, both with SFT and LAIF, using the prompts from the ShareGPT dataset [Chiang et al., 2023]. We use SFT demonstrations from three teacher language models, namely, GPT-3.5, GPT-4, and Claude, and use two strong language models as critics, GPT-4 and Claude, for collecting AI feedback. We evaluate our fine-tuned models using AlpacaEval [Li et al., 2023]. The experimental setup is described in further detail in Section 4. In our experiments, we find that two conditions are necessary for LAIF to significantly outperform SFT: (a) *a sufficiently strong pre-trained base model* and, (b) *a capability mismatch between the teacher used for the SFT data collection and the critic used for collecting AI feedback.* The latter condition has surprising consequences: if the target completions are sufficiently performant and capability gap between models used for SFT and AI feedback is minimal, then simply doing SFT can suffice. We observe this in Figure 1, where SFT on the completions generated by GPT-4 outperforms GPT-3.5 SFT + GPT-4 feedback. This suggests that LAIF from a strong critic may be compensating for a weak teacher in popular SFT datasets like

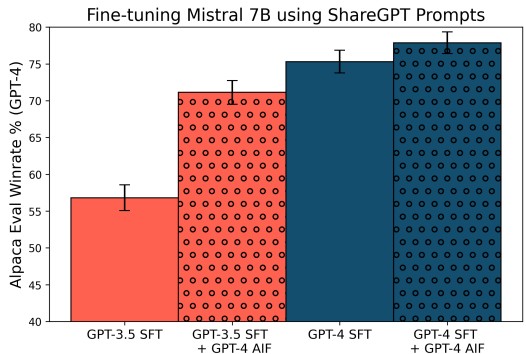

Figure 1: **Supervised fine-tuning (SFT) on strong teachers can accounts for improvements from learning from AI feedback (LAIF).** LAIF from strong models such as GPT-4 can result in substantially better instruction-following LLMs than supervised SFT alone on popular datasets such as ShareGPT [Chiang et al., 2023] constructed using GPT-3.5 completions (GPT-3.5 SFT + GPT-4 AIF). However, simply performing SFT on completions from GPT-4 can result in a better model (GPT-4 SFT), suggesting that improvement in performance from LAIF is partly because the default ShareGPT completions are from a weak teacher (GPT-3.5). Furthermore, LAIF (GPT-4 SFT + GPT-4 AIF) does not result in a significantly better model compared to GPT-4 SFT alone.

ShareGPT that are generated using GPT-3.5, and we may be overestimating the effectiveness of AI feedback. We further try to analyze these findings in a more principled way in Section 5, and provide a possible mechanistic intuition in a simplified bandit setting for why SFT might perform comparably or even outperform LAIF.

Finally, Section 6 provides practical suggestions and discussions based on the experimental and theoretical insights in the previous sections. First, one should account for the distribution of completions in instruction-tuning datasets [Taori et al., 2023, Chiang et al., 2023, Ding et al., 2023] before evaluating AI feedback based methods. Further, we should consider updated instruction-tuning datasets created using more recent state-of-the-art LLMs, like GPT-4 (as of February 2024). Second, while our experiments suggest that SFT on strong teacher completions can be just as effective as LAIF, we caution readers into generalizing these findings to RLHF setting and highlight why preference-based RL might be superior when collecting human feedback due to inherent cognitive load of collecting demonstrations. Third, our experiments further support the importance of pre-training for effective instruction-following, both in terms of absolute performance and also in its ability to fine-tune from AI feedback. Finally, we also provide possible ways to improve the effectiveness of AI feedback and hypothesize how AI feedback could improve over SFT over target distributions from strong teacher LLMs. Overall, AI feedback has several desirable features that make it a compelling avenue for scalable oversight and effective alignment, but current evaluation may be overestimating the

significance of improvements in instruction-following for open-source LLMs and we hope that this paper will inspire investigation into more effective LAIF approaches.

## 2 Related Work

Modern language models rely on a two-stage pipeline that first learns representations with large-scale unsupervised learning and then adapts, or fine-tunes, those representations to the task of interest; this strategy initially proved effective for isolated word embeddings [Collobert et al., 2011] and more recently has been adopted for sequence-level representations [Devlin et al., 2019, Radford and Narasimhan, 2018]. Virtually all of the most powerful language models are pre-trained with a simple maximum likelihood objective over a large, unsupervised dataset [Radford and Narasimhan, 2018, Radford et al., 2019, Brown et al., 2020, Touvron et al., 2023a,b]. Owing to the drastically smaller computational demands of fine-tuning, considerably more diversity exists in the objectives used for fine-tuning, including supervised objectives such as imitation [Collobert et al., 2011, Devlin et al., 2019, Radford et al., 2019, Raffel et al., 2020, Taori et al., 2023] and ranking losses [Yuan et al., 2023] as well as reinforcement learning from heuristic rewards [Paulus et al., 2018], learned rewards trained from human reward scores [Böhm et al., 2019] or preference annotations [Ziegler et al., 2020, Ouyang et al., 2022, Bai et al., 2022, Rafailov et al., 2023].

Simultaneous with the development of new fine-tuning objectives, several works have focused on the source of data used for fine-tuning. While human-annotated data has been widely used [Collobert et al., 2011, Devlin et al., 2019, Radford and Narasimhan, 2018, Ouyang et al., 2022, Conover et al., 2023], more scalable synthetic data sources have recently been explored, using language models themselves to generate some or all of the fine-tuning data. Bai et al. [2022], Chiang et al. [2023], Taori et al. [2023], Teknium [2023] show that outputs generated by a high-quality language model can be distilled into a smaller language model through supervised imitation. Bai et al. [2022] also show that a large language model can effectively generate preferences over model samples, which are ultimately used to train a reward model used for further RL-based fine-tuning, which is referred to as learning from AI feedback (LAIF). Tunstall et al. [2023], Ivison et al. [2023], Intel [2023] further explore using model-generated datasets with various combinations of supervised and reinforcement learning-based fine-tuning objectives, suggesting that LAIF can produce improvements over purely supervised fine-tuning from AI feedback. However, notably, commonly-used datasets for supervised training [Taori et al., 2023, Ding et al., 2023, Geng et al., 2023] contain substantial amounts of data generated by GPT-3.5 [OpenAI, 2023], while the more recently-collected AI datasets for LAIF contain annotations generated by the stronger GPT-4 model [Ding et al., 2023]. The ramifications of this discrepancy, and how the effectiveness of LAIF is affected by the teacher and critic quality is the focus of this work.

## 3 Algorithmic Overview of LLM Fine-tuning

The dominant approaches to fine-tuning LLMs as general dialogue agents are supervised fine-tuning (SFT) and preference-based reinforcement learning from either human or AI feedback (RLHF or LAIF, respectively).

**Supervised fine-tuning.** Consider a dataset $\mathcal{D}_{\text{SFT}} = \{x, y\}$, where $x$ is one or more turns of dialogue history[1] and $y$ is the target response (a sequence of tokens) for this dialogue history, which the model is trained to generate. Specifically, an autoregressive language model $p_\theta$ parameterized by $\theta$ is trained to minimize the negative log likelihood of the responses, given the histories:

$$\mathcal{L}_{\text{SFT}} = -\mathbb{E}_{(x,y)\sim\mathcal{D}} \log p_\theta(y \mid x) \tag{1}$$

The targets $y$ may be written by either humans or strong LLM teachers.

**RLHF & LAIF.** Motivated by the difficulty of gathering high-quality target responses $y$ at scale from humans, an alternative to purely supervised fine-tuning collects *preference* annotations over pairs of candidate responses $y, y'$ to a dialogue history $x$. Typically, a model fine-tuned with SFT is used to generate the pair of responses,[2] and humans for RLHF or strong LLM critics for LAIF annotate

---

[1]In many cases, $x$ contains a 'system prompt' prefix that is used to steer the high-level behaviors of the language model across dialogues.

[2]Or larger sets, but for simplicity, we consider pairs here.

which better responds to the input according to some set of criteria. We refer to the preferred response as $y_w$ and the dispreferred response as $y_l$, producing a dataset of the form $\mathcal{D}_p = \{x^i, y_w^i, y_l^i\}$. Using a theoretical model of human discrete choice such as the Bradley-Terry model [Bradley and Terry, 1952], which relates discrete choices to implicit goodness scores of the underlying options, we can train a reward model with maximum likelihood using this preference data. For the Bradley-Terry model, the reward modeling loss is:

$$\mathcal{L}_{\text{BT}} = -\mathbb{E}_{(x, y_w, y_l) \sim \mathcal{D}_p} \log p_\phi(y_w \succ y_l \mid x) \tag{2}$$

$$= -\mathbb{E}_{(x, y_w, y_l) \sim \mathcal{D}_p} \log \sigma \left( r_\phi(x, y_w) - r_\phi(x, y_l) \right) \tag{3}$$

where $\sigma$ denotes the sigmoid, and $r_\phi$ denotes the reward function with parameters $\phi$ to fit the goodness score of a completion $y$ for an input prompt $x$. The equality between the two RHS expressions is due to choice of the Bradley-Terry model. Using a reward model trained with this objective, the final step is to fine-tune a generative language model policy $\pi_\theta$ that generates high-scoring responses. Typically, $\pi_\theta$ is initialized from a model trained with SFT $\pi_{\text{SFT}}$, and regularized to keep the KL divergence between the $\pi_\theta$ and $\pi_{\text{SFT}}$ small, giving the final policy search objective

$$\pi^* = \max_{\pi_\theta} \mathbb{E}_{x \sim \mathcal{D}, y \sim \pi_\theta(\cdot \mid x)} \left[ r_\phi(x, y) - \beta \text{KL} \left( \pi_\theta(\cdot \mid x), \pi_{\text{SFT}}(\cdot \mid x) \right) \right], \tag{4}$$

where $\mathcal{D}$ is a dataset of unlabeled prompts and $\beta$ controls the strength of the KL regularization.

The original form of RLHF for language models [Böhm et al., 2019, Ziegler et al., 2020] used relatively expensive online reinforcement learning algorithms such as A2C [Mnih et al., 2016] or PPO [Schulman et al., 2017]. More recently, Rafailov et al. [2023] show that the optimal policy for the learned reward can be extracted in closed form, essentially skipping the need to perform iterative, approximate policy learning. The resulting algorithm, direct preference optimization (DPO), is simpler to tune and less computationally demanding than prior methods, while optimizing the same objective. We therefore use DPO as the algorithm for LAIF in our experiments. The DPO loss for the language model policy $\pi_\theta$ is

$$\mathcal{L}_{\text{DPO}} = -\mathbb{E}_{x, y_w, y_l \sim \mathcal{D}_p} \log \sigma \left( \beta \log \frac{\pi_\theta(y_w \mid x)}{\pi_{\text{SFT}}(y_w \mid x)} - \beta \log \frac{\pi_\theta(y_l \mid x)}{\pi_{\text{SFT}}(y_l \mid x)} \right), \tag{5}$$

which trains the language model policy $\pi_\theta$ directly, without the need to first train a separate reward model $r_\phi$.

## 4 Experiments

In light of increasing usage of LAIF, the primary question we want to investigate is whether preference-based RL from AI feedback, produced by blackbox LLMs, like GPT-4 is more effective at aligning language models compared to simple supervised fine-tuning on completions from these LLMs. To this end, we first explain the experimental setup for comparing LAIF and SFT in 4.1. With this experimental setup, we observe in 4.2 that LAIF can result in substantially more capable models than those obtained by instruction-tuning on current public SFT datasets. However, we also find that using a stronger teacher for SFT consistently matches or outperforms LAIF. This surprising result spurs deeper investigation into effectiveness of LAIF, where we find that using a stronger SFT distribution also limits any further gains from AI feedback.

### 4.1 LAIF Setup and Data Scaling for SFT

**Datasets & Models.** To systematically compare SFT and LAIF on their effectiveness for instruction following, it is important to control the dataset of instructions used to train both the methods. To this end, we fix the dataset of prompts to be single-turn instructions derived from ShareGPT [Chiang et al., 2023]. We consider a variety of performant pre-trained LLMs for our experiments: Llama [Touvron et al., 2023a,b], Mistral [Jiang et al., 2023] and Mixtral [Jiang et al., 2024], Yi [01.AI, 2023] and DeepSeek [Bi et al., 2024].

**Terminology.** We will use the strong LLMs like GPT-4 in three different roles: as a *teacher* where the LLM generates a target completion for an instruction, as a *critic* where the LLM gives feedback on which completion it prefers for a given instruction, and similarly as an *evaluator* where it labels the preferred completion on a held out set of instructions.

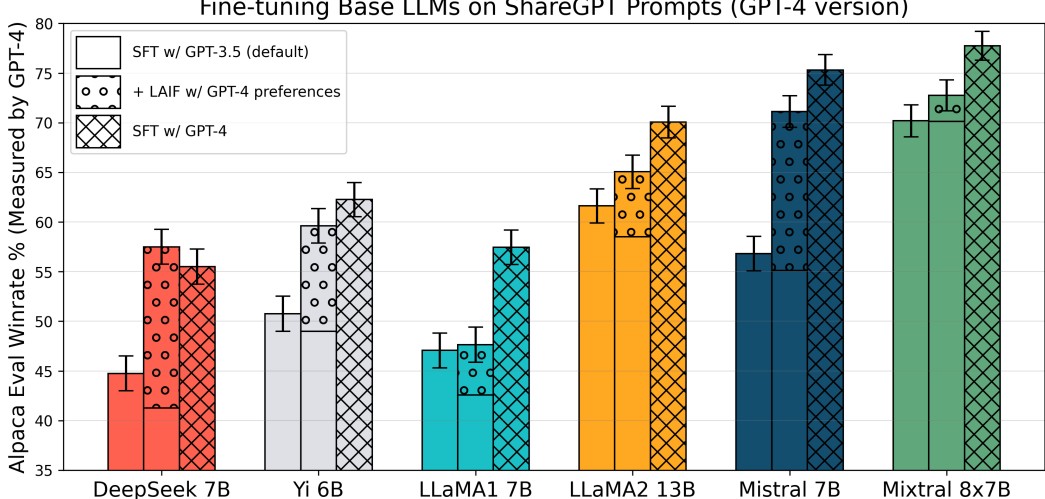

Figure 2: **SFT can perform comparably or better than LAIF across various model sizes and classes.** The same set of prompts are used for all three settings and for each model, and the oracle LLM either generates a completion (*for SFT*) or a preference label on two completions (*for LAIF*). For LAIF, SFT is an important precursor, so we SFT on 10% of the total prompts, and LAIF is done on the remaining 90%. For the other settings, the full set of prompts are used for SFT. While LAIF improves the performance compared to SFT on the default ShareGPT completions, SFT on GPT-4 completions consistently matches or outperforms LAIF.

**Setup.** When fine-tuning these pre-trained models with SFT, we consider the default target distribution in ShareGPT, which is sampled from GPT-3.5 as the teacher. We also generate completions from GPT-4 (`gpt-4-0314`) and Claude (`claude-v1`) for the same set of instructions For LAIF, we consider the following pipeline:

1. Split the training prompts for SFT and LAIF. Fine-tune the base model on completions from teacher $\mathcal{M}_{\text{teacher}}$ to minimize $\mathcal{L}_{\text{SFT}}$.
2. For prompts in the AIF split, create a dataset by sampling two completions for every prompt and label the preferred completion using $\mathcal{M}_{\text{critic}}$ to create $\mathcal{D}_p$.
3. Fine-tune the instruction-tuned model using $\mathcal{L}_{\text{DPO}}$.

**Prompt split for SFT and AIF**. For a controlled study comparing AIF and SFT, one would ideally compare the performance of a base LLM fine-tuned to minimize $\mathcal{L}_{\text{DPO}}$ on a preference dataset $\mathcal{D}_p$ containing *all* prompts, to that of a base LLM fine-tuned to minimize $\mathcal{L}_{\text{SFT}}$ on *all* prompts. However, prior works have found instruction-tuning to be a necessary precursor for successful RL fine-tuning [Ouyang et al., 2022, Touvron et al., 2023b, Rafailov et al., 2023, Tunstall et al., 2023], which motivates the inclusion of first step in the pipeline. However, we still want to predominantly evaluate the effectiveness of AIF, therefore we want to use majority of the prompts for AIF. We observe in Figure 6 that when we evaluate the SFT-only instruction following performance of Llama7B as instruction prompts are scaled, performance improves rapidly as instruction-tuning data is increased initially, but the improvement is minimal as the SFT data is further increased. This also corroborates observations in prior work [Zhou et al., 2023, Touvron et al., 2023b]. Therefore, we use 10% of the available prompts for the SFT stage and the rest of them to generate the AIF dataset. This provides a strong initialization for LAIF while still leaving a large disjoint set of prompts to use for training on AI feedback.

**Response Pairs for AI Feedback**. To sample response pairs for AI feedback, we construct preference pairs by using one completion from $\mathcal{M}_{\text{teacher}}$ and one completion from $\pi_{\text{SFT}}$. In a typical RL pipeline, the preference dataset is generated by sampling two completions from $\pi_{\text{SFT}}$ for every prompt. However, we find that using the completions from $\mathcal{M}_{\text{teacher}}$ as one of the inputs in the preference pair results in better performing models. Thus, we evaluate LAIF in the most favorable possible circumstances by using this scheme to construct preference pairs.

**Evaluation protocol.** We evaluate all the instruction-tuned models using AlpacaEval [Li et al., 2023], where an evaluator model $\mathcal{M}_{\text{eval}}$ compares the outputs of the current model with the outputs of instruction-tuned GPT-3 (`text-davinci-003`), and these preference labels are averaged over 805 instructions to give a win rate for the model. This evaluation protocol essentially compares how well a given model adheres to the preferences of $\mathcal{M}_{\text{eval}}$. *The evaluation questions in AlpacaEval have no overlap with our training prompts from ShareGPT.* To minimize discrepancy between the critic preferences and evaluation preferences, we will use the same model for both, that is $\mathcal{M}_{\text{eval}} = \mathcal{M}_{\text{critic}}$, reducing concerns about overoptimization [Gao et al., 2022]. Further, we also use the same prompt template for labeling training preferences and computing evaluation win rates. For all the experiments, we restrict $\mathcal{M}_{\text{critic}}$ to be GPT-4 or Claude, and the prompt template for them is shown in Appendix B.

Further details about dataset construction, hyperparameters for SFT and DPO can be found in Appendix A. We use DPO for all the experiments in this section for its computational simplicity and effectiveness, but we also provide additional PPO based experiments in Appendix D.

## 4.2 Fine-tuning under Different Target Distributions

**Comparing ShareGPT SFT and LAIF.** First, we compare fine-tuning various base LLMs with SFT on the default target completions in ShareGPT, i.e, $\mathcal{M}_{\text{teacher}} = $ GPT-3.5. We use $\mathcal{M}_{\text{critic}} = $ GPT-4 in Figure 2 and $\mathcal{M}_{\text{critic}} = $ Claude in Figure 3 for AI feedback. We find that LAIF can substantially improve instruction following performance over instruction-tuning on GPT-3.5 across all base models. For both GPT-4 and Claude feedback, we find that instruction-following performance can improve by over 10% in absolute win rate. The only exception is the LAIF with GPT-4 feedback for Llama 7B where the improvement is far more modest; we investigate this outlier later in greater depth. **Using completions from $\mathcal{M}_{\text{critic}}$ for SFT.** While LAIF provides substantial improvement over SFT, there is an important discrepancy: The default SFT samples in ShareGPT are from a rel-

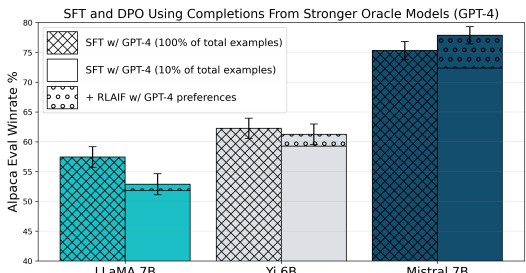

Figure 3: **We make a similar observation that SFT performs comparably to LAIF when Claude as is used as an oracle**. LAIF with AI feedback from does not significantly outperform SFT on Claude completions, and the performance improvement from LAIF is explained by the use of a weaker SFT target distribution (GPT-3.5). The results for effectiveness of SFT may apply more generally to strong LLMs beyond GPT-4.

atively weaker model, while the AI feedback is collected from the stronger model $\mathcal{M}_{\text{critic}}$ (GPT-4 or Claude). What if we use the stronger LLM $\mathcal{M}_{\text{critic}}$ as a teacher? We evaluate LLMs fine-tuned with SFT on GPT-4 completions in Figure 2 and on Claude completions in Figure 3, and we find that these SFT models *consistently outperform or match LAIF* for all base models. This strongly suggests the performance of SFT models was limited by the quality of the instruction-tuning data; accounting for the SFT distribution is important when evaluating models using AI feedback.

We note that this phenomenon is not self-evident apriori. One would expect that a model trained with an explicitly preference-based objective, in the favorable setting where the same ranking model is used both as critic and evaluator, would perform better on the fundamentally preference-based AlpacaEval evaluation. Instead, we find that a model trained on SFT, where the objective is less congruent with the evaluation, performs comparably or better. We offer possible explanations to this in Section 5.

**What if we use completions from $\mathcal{M}_{\text{critic}}$ in LAIF too?** A natural question to ask is how does LAIF performs when we use completions from $\mathcal{M}_{\text{critic}}$ during the SFT step of LAIF (ie Step 1). In this experiment, the completions from $\mathcal{M}_{\text{critic}}$ on 10% of the prompts are used to

Figure 4: **SFT on strong distribution minimizes any improvements from LAIF.** We consider LAIF starting with GPT-4 completions as the target distribution for SFT in the first step, and also to generate preference labels for RL. We find surprisingly minimal improvements in performance when compared to SFT on completions sampled from GPT-4 for all prompts. This experiment suggests that SFT on the target completions from the strong critic can minimize any further gains from LAIF.

train the SFT model $\pi_{\text{SFT}}$, and pairs of $\mathcal{M}_{\text{critic}}$ and $\pi_{\text{SFT}}$ completions are used for the rest of the prompts to form the AI feedback data. These preference pairs are labeled by the same $\mathcal{M}_{\text{critic}}$. While the SFT on 10% $\mathcal{M}_{\text{oracle}}$ completions is substantially better than before, we find that **benefits of the complete LAIF pipeline are dramatically diminished, providing little to no benefit over SFT alone, when $\mathcal{M}_{\text{critic}}$ is used as teacher too**. This finding holds when using either GPT-4 in Figure 4 and Claude in Figure 7 as the teacher/critic and for various base LLMs.

The results in this section suggest the following hypothesis: AI feedback is effective when there is a discrepancy between the SFT distribution and the evaluator, that is, $\mathcal{M}_{\text{teacher}}$ is worse than $\mathcal{M}_{\text{critic}}$. This is implied by the observation that LAIF pipeline, with SFT targets generated by $\mathcal{M}_{\text{critic}}$, performs similarly or worse than simple SFT with $\mathcal{M}_{\text{critic}}$ as the teacher too. But, the same LAIF pipeline substantially improves the model when $\mathcal{M}_{\text{teacher}}$ is worse than $\mathcal{M}_{\text{critic}}$ (i.e., GPT-3.5 as the teacher and GPT-4 as the critic).

## 5  Possible Mechanistic Explanations for the Ineffectiveness of LAIF

The experiments in Section 4 suggest a surprising result that SFT on the *right* target distribution can match or even outperform LAIF, especially when one considers that the preferences used during training and evaluation are generated by the *same model* and using the *same prompting template*. The LAIF objective of maximizing the (implicit) reward implied by the training preferences is closely aligned with the evaluation objective of generating the preferred completion. Why then would a model fine-tuned to maximize the log-likelihood on potentially suboptimal completions outperform LAIF on this evaluation? Note, even the completions sampled directly from $\mathcal{M}_{\text{critic}}$ are not necessarily optimal for the reward function implied by the evaluator, and optimizing $\mathcal{L}_{\text{SFT}}$ on a finite dataset does not guarantee the recovery of the optimal policy [Ross et al., 2011]. In this section, we consider some empirically motivated hypotheses for why LAIF might be ineffective compared to SFT in our current experiments.

**Current base LLMs are insufficiently responsive to AI feedback**. Most base LLMs improve substantially from GPT-4 feedback when instruction-tuned on GPT-3.5 completions in ShareGPT in Figure 1, except for Llama models. Particularly for Llama 7B, where the absolute performance is also low, LAIF seems to be relatively ineffective at improving performance (see both Figure 1 and Figure 4). There are two possible reasons: either the preference dataset generated by Llama 7B (after SFT) is not informative enough to learn from, or the base model has limited ability to improve from RL fine-tuning. We consider the following experiment to resolve this: We fine-tune the Llama 7B SFT model with DPO on the preference dataset generated from samples from Mistral 7B, fine-tuned with SFT on GPT-3.5. Similarly, we fine-tune Mistral 7B with SFT on the preference dataset using samples from Llama 7B. Surprisingly, we find in Table 1 that performance after swapping preference datasets is close to the original performance of the models. That is, Mistral 7B performs nearly as well training on preferences generated by Llama 7B as it would training on its own preferences.

This suggests that the *performance of a model when learning from preferences may be tied to the base LLM itself, rather than exclusively the exploration performed during the sampling of the preference data responses*. DPO fine-tunes a LLM by a solving a classification problem, where the model learns to discern preferred completions from dispreferred completions based on preference labels. Improvements in instruction-following performance when solving this classification performance may be tied to the underlying representation space of the models. Based on the fact that LAIF on Llama 7B does not im-

|         |            | Preference Distribution $\mathcal{D}_p$ | |
|         |            | Llama 7B | Mistral 7B |
|---------|------------|----------|------------|
| **Base** | Llama 7B   | 45.0 (1.76) | 47.6 (1.76) |
| **LLM**  | Mistral 7B | 68.5 (1.64) | 71.1 (1.60) |

Table 1: LAIF with preference data responses sampled from a different model than the base model being fine-tuned. We find that the final performance after fine-tuning is affected more by the choice of the base LLM, as Mistral 7B reaches a similar performance when fine-tuning on preferences over Llama 7B responses, whereas Llama7B does not improve significantly when trained on preferences over responses generated by Mistral 7B.

prove substantially over SFT on GPT-3.5 responses alone, a possible hypothesis is that all the current base LLMs may provide limited improvements over GPT-4 SFT because the representation space does not improve further for instruction-following by improving the RLHF objective, even though the preference data with *right* base LLM can lead to more performant instruction-following model than SFT.

**Completions sampled from SFT models are substantially poorer than completions sampled from** $\mathcal{M}_{\mathbf{oracle}}$. Oracle models like GPT-4 and Claude are high-quality instruction-following LLMs; thus, the completions generated by these LLMs would be higher-quality than the completions generated by SFT models considered in our experiments. Assuming that $\mathcal{M}_{\mathrm{critic}}$ completions rank sufficiently higher than $\pi_{\mathrm{SFT}}$ completions according to $\mathcal{M}_{\mathrm{eval}}$'s ranking over all possible responses, would it better be to imitate the completions from $\mathcal{M}_{\mathrm{critic}}$ or improve the LLM via via LAIF?

Consider an illustrative bandit problem in Figure 5 with a discrete action space of size 100, where the model completions from a strong teacher distribution completions (*hyphenated black*) rank around the 80th percentile under some 'true' reward function (*light dashed line*), whereas the model completions from a weak student $\pi_s$ (*hyphenated blue*) rank around the 40th percentile. The SFT policy (*red*) is learned by minimizing $\mathcal{L}_{\mathrm{SFT}}$ on 500,000 actions sampled from the teacher distribution. For LAIF (*yellow*), we sample 500,000 pairs of completions and label preferences on each pair of completions using the true reward. We fit a reward function by minimizing $\mathcal{L}_{\mathrm{BT}}$ and compute the optimal LAIF policy analytically as $\pi_{r_\phi}(a) \propto \pi_s(a)\exp(r_\phi(a)/\beta)$, where $r_\phi(\cdot)$ is the learned reward function (*light solid line*), $\pi_s$ denotes the initial student distribution, and the temperature $\beta = 0.1$. We find that while the SFT policy has a higher variance than the LAIF policy, it has a higher expected reward by simply imitating the actions sampled from the teacher. On the other hand, we observe that while LAIF policy improves the student policy, it is still substantially suboptimal.

We make several observations about this result. First, the student fails to outperform the teacher due to inadequate exploration; the preference dataset simply does not contain any high-quality responses to reinforce. Therefore, even though the teacher policy is suboptimal under the evaluation reward, imitating the teacher still outperforms LAIF with preferences annotated by $\mathcal{M}_{\mathrm{critic}}$. Finally, while theoretically one could improve the LAIF policy further by setting a lower $\beta$, in practice, we are limited by the imperfect learned reward which can be exploited as the temperature is decreased.

However, this experiment also suggests that LAIF may be able to learn a policy that improves over the SFT policy *if* the samples from the student policy include actions above the 80th percentile. Further, with the small temperature values used in practice (e.g., Rafailov et al. [2023]), the LAIF policy may have much lower variance than policies trained with SFT. Nonetheless, if the mode is located around high reward completions, the LAIF policy can have higher expected return. In contrast, the SFT policy may end up with higher variance by virtue of minimizing an imitation loss and thus lower expected reward.[3]

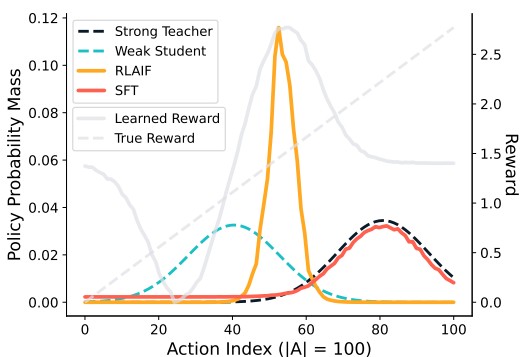

Figure 5: **Fine-tuning a weak student with LAIF underperforms relative to SFT on a strong teacher** in a synthetic bandit problem with 100 possible actions. We assume the completions from the teacher (*black*) rank relatively highly (centered around 80th percentile). The improvements in LAIF (*yellow*) are limited because the actions sampled for labeling preferences are tied to the initial student distribution (*blue*). In this scenario, where the teacher distribution is sufficiently stronger than the student distribution, simple SFT on the teacher's samples (*red*) may be more effective than LAIF on samples from a weak student. The actions are sorted by their **true reward**, which is used to generate a teacher labeled preference dataset over samples from the student.

## 6  Going Forward

In light of our results, we suggest several points of consideration for future implementations of LAIF.

**Improve Datasets for Instruction-Tuning**. In Section 4.2, we showed how the performance gains of LAIF over the SFT might be partially attributed to mismatch between the evaluator ($\mathcal{M}_{\mathrm{critic}}$) and the model used to collect the SFT data ($\mathcal{M}_{\mathrm{teacher}}$). Just like ShareGPT [Chiang et al., 2023], numerous other AI-generated instruction-tuning SFT datasets such as Alpaca [Taori et al., 2023], Self-Instruct [Wang et al., 2022], UltraChat [Ding et al., 2023] are collected using relatively weaker

---

[3]However, the relationship between policy variance and quality depends on whether evaluation is performed by expected reward (for which a single very bad response can 'spoil' an otherwise strong model) or win rates (for which the contribution of a single very bad response to overall model scoring is limited).

GPT-3.5 models. It is important to consider and account for the instruction-tuning distribution when studying RLHF and LAIF, as our experiments show that it can drastically change the conclusions. The next generation of distilled models can also benefit from improved instruction-tuning datasets sampled from more recent and performant LLMs. Further, versioning and continual updating of these datasets will reduce drift in quality between SFT and AI feedback datasets. Some recent datasets are collecting better instruction-tuning datasets, such as the GPT4-LLM dataset Peng et al. [2023], also demonstrating better instruction-tuned models. More such datasets would not only result in stronger SFT fine-tuned models but would also help the community study the new and existing AI feedback mechanisms, which often rely on state-of-the-art models as evaluators, in a fairer setting.

**Implications for RLHF.** As motivated in the introduction and shown by our experiments, using LLMs for generating completions or providing feedback in the RLHF pipeline introduces new considerations for data collection and algorithms to use for optimization, specifically that SFT on the right target distribution can be much more performant. While one may be tempted to conclude that SFT may suffice in case of human completions and human feedback, the performance gains of RLHF over SFT are well documented Ouyang et al. [2022], Bai et al. [2022], Rafailov et al. [2023]. We postulate that this dichotomy is due to how data collection works for RLHF and LAIF. In RLHF, the completions for the SFT stage are collected by humans, whereas most LAIF instruction tuning SFT datasets use LLMs for completion generation.[4] Humans, when asked to generate completions, may not be sufficiently incentivized to generate high-quality completions for instructions, especially when collecting data at scale. For example, writing detailed and insightful answers for every question may be a demanding task, and humans may also not be sufficiently informed to write such answers. However, humans may implicitly prefer longer detailed answers [Singhal et al., 2023], and thus, when giving feedback on completions for RLHF, may encourage such behavior. This discrepancy between the quality of human completions and what humans prefer may explain the effectiveness of RLHF over SFT.

Additionally, in terms of data collection efficiency for *human* feedback, it is easier for humans to generate preference labels as opposed to generating completions. However, in the case *AI* feedback, the computational cost of collecting preferences may be higher than generating completions as the length of the context for collecting preferences is usually considerably longer than total length (context and generation) of the output while collecting completions.

**Is AI Feedback Simply Ineffective?** In this paper, our limited claim is that adapting preference-based learning from the RLHF framework by replacing humans with AI might lead to sub-optimal outcomes from the perspectives of both downstream performance and cost-effectiveness. Our experiments suggest that the performance gains of LAIF depend upon several factors that can impact final performance in non-obvious ways, such as the base pre-trained model size and class, the SFT data distribution, and the quality of the exploration performed in the generated responses in the preference data. Nonetheless, our discussion in Section 5 suggests that if (1) preference datasets contain adequate exploration and (2) AI feedback provides high-quality preference labels, learning from AI feedback may be a promising and scalable approach to training capable instruction-following models. Moreover, alignment can have broader goals than improving instruction following abilities [Ouyang et al., 2022, Bai et al., 2022], in which case AI feedback can be leveraged for inducing safe and desirable behavior in a cheap and automated fashion.

## 7 Conclusion

In this paper, we critically evaluate the current prevalent paradigm of LAIF, which replaces humans with strong LLM annotators for labeling feedback. We observe in our experiments that current gains observed by open-source LAIF models might be an artifact of the capability mismatch between the models generating SFT data and the AI feedback, and effectiveness of LAIF for strong-instruction following LLMs may be overestimated. We also showed that LAIF gains do not translate universally across models, evaluators, and oracle models. We our analysis is limited, we provide some mechanistic insight into this behavior via a simplified bandit setting. Finally, we provide some practical suggestions and discuss the implications of these results for future LAIF and RLHF research. Concretely, we recommend periodic versioning and regular updates to AI-generated instruction fine-tuning datasets with the release of more powerful language models. More design and investigation is needed to extract the best possible performance from LAIF methods.

---

[4]There are some exceptions to this such as the OpenAssistant Conversations Dataset Köpf et al. [2023].

# 8 Acknowledgements

We would like to thank Adrien Gaidon, Rares Ambrus, Achal Dave, Jean Mercat and Igor Vasiljevic for their insight and advice during the early development of this work.

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

# A    Detailed Experimental Setup and Hyperparameters

## A.1    Data Processing

We use ShareGPT [Chiang et al., 2023] for all our experiments. We filter the prompt to only single-turn conversations, which gives us about 46,000 (instruction, completion) pairs, where the completions are sampled from GPT-3.5. To preprocess the dataset, we use truncate every prompt to a maximum of 256 tokens, and truncate the completion such that the total length of the prompt and completion combined does not exceed 512 tokens. We use the following template for all our instruction-tuning experiments:

```
\n\nHuman: {instruction}\n\nAssistant: {completion}<eos>
```

## A.2    Training Settings and Hyperparameters

For SFT runs, we train the models on 9 epochs and evaluate every 3 epochs. From here, we select the best checkpoint to report. We use a batch size of 8 and conduct a hyperparameter sweep for learning rate across {1e–7, 5e–7, 1e–6}. We found some models to converge faster than others. For instance, for the Mistral-7B and Mixtral-8x7B models, converged within 3 epochs, so we conducted our hyperparameter sweep over 3 epochs and evaluated the model every epoch.

For DPO, we select the best SFT checkpoint and train on top of it. We do DPO training for 3 epochs and evaluate every epoch. Learning rate for DPO runs is fixed at 5e–7 and beta at 0.05 for all models. We have also released the code and datasets for reproducing the experiments. Training was done on A100 80GB instances and took around 1 hour per epoch for a 7B model when trained on 100% of the training examples.

# B    Preference Labeling and Evaluation Prompt Templates

Prompt template used by GPT-4 in AlpacaEval [Li et al., 2023]:

```
<|im_start|>system
You are a helpful instruction-following assistant.
<|im_end|>
<|im_start|>user
Select the output (a) or (b) that best matches the given instruction. \
Choose your preferred output, which can be subjective. Your answer should \
ONLY contain: Output (a) or Output (b). Here's an example:

# Example:
## Instruction:
Give a description of the following job: "ophthalmologist"

## Output (a):
An ophthalmologist is a medical doctor who specializes in the diagnosis \
and treatment of eye diseases and conditions.

## Output (b):
An ophthalmologist is a medical doctor who pokes and prods at your eyes \
while asking you to read letters from a chart.

## Which is best, Output (a) or Output (b)?
Output (a)

Here the answer is Output (a) because it provides a comprehensive and \
accurate description of the job of an ophthalmologist. In contrast, \
output (b) is more of a joke.

# Task:
```

```
Now is the real task, do not explain your answer, just say Output (a) \
or Output (b).

## Instruction:
{instruction}

## Output (a):
{output_1}

## Output (b):
{output_2}

## Which is best, Output (a) or Output (b)?
<|im_end|>
```

Prompt template for Claude:

```
Human: Select the output (a) or (b) that best matches the given instruction. \
Choose your preferred output, which can be subjective. Your answer should \
ONLY contain: Output (a) or Output (b). Here's an example:

# Example:
## Instruction:
Give a description of the following job: "ophthalmologist"

## Output (a):
An ophthalmologist is a medical doctor who specializes in the diagnosis \
and treatment of eye diseases and conditions.

## Output (b):
An ophthalmologist is a medical doctor who pokes and prods at your eyes \
while asking you to read letters from a chart.

## Which is best, Output (a) or Output (b)?
Output (a)

Here the answer is Output (a) because it provides a comprehensive and accurate \
description of the job of an ophthalmologist. In contrast, output (b) is more \
of a joke. Now is the real task, remember to only include Output (a) or Output (b) \
in your answer, not the explanation.

# Task:
Now is the real task, do not explain your answer, just say Output (a) or Output (b).

## Instruction:
{instruction}

## Output (a):
{output_1}

## Output (b):
{output_2}

## Which is best, Output (a) or Output (b)?

Assistant:
```

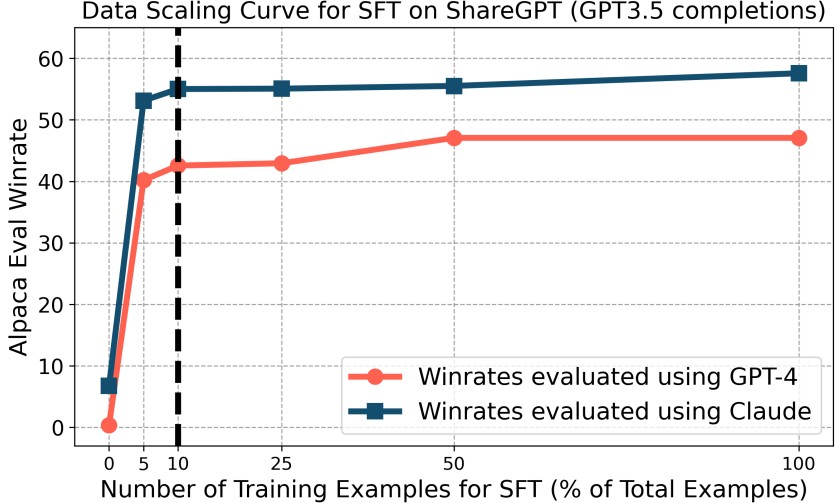

Figure 6: The performance improvements from increasing the number of training points SFT on 100% of the training prompts yields minimal improvements over SFT on 10% of the training prompts. Hence, for our LAIF setting, we first perform SFT on only 10% of the training examples, and we use the remaining for LAIF.

## C  Additional Results

In Figure 4, we evaluate the performance of various models when both completions and AI feedback is generated by GPT-4. We conduct a similar experiment with Claude in Figure 7, where the a Llama 7B is trained is fine-tuned with SFT on 10% of the prompts, followed by LAIF on 90% of the prompts with AI feedback from Claude. We find that a model instruction-tuned using LAIF underperforms a model fine-tuned using SFT on Claude completions on all prompts, replicating the observation in a similar setup with GPT-4.

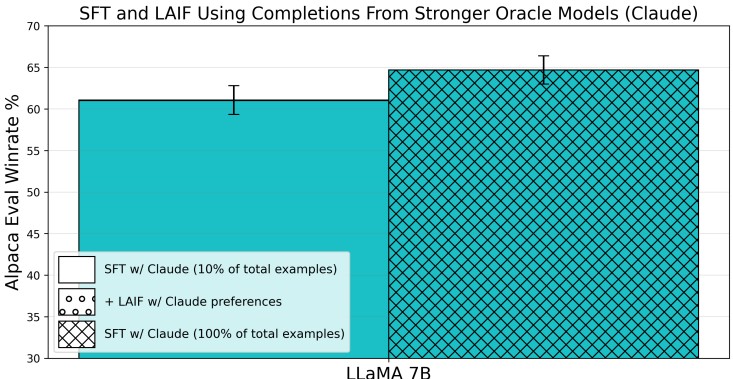

Figure 7: **SFT on strong distribution minimizes any improvements from LAIF.** We consider LAIF starting with Claude completions as the target distribution for SFT in the first step, and also to generate preference labels for RL. We find SFT on Claude completions for all prompts (*SFT 100%*) outperforms LAIF, where we observe no improvement from LAIF. This experiment suggests that SFT with the right target distribution can potentially minimize any further gains from LAIF, even when starting with SFT on completions from the oracle model.

## D  PPO-based RLAIF Experiments

In this section, we present attempt to verify if our findings also hold in the RLAIF setting, i.e., where the policy is trained using an online reinforcement learning algorithm such as PPO, rather than an

offline equivalent such as DPO. The standard RLAIF pipeline has three stages: 1.) SFT fine-tuning, 2.) reward training, and 3.) PPO fine-tuning. Following the same protocol as Bai et al. [2022] and **?**, we adapt the standard RLHF pipeline [Christiano et al., 2017] for AIF, by replacing humans with a strong AI model, both as a teacher for SFT fine-tuning data collection and as a critic for preference collection for reward modeling. Next, we discuss our experimental setup in detail.

**Experimental Setup**

**Data:** Similar to our DPO-AIF experiments, we use the 10% prompts and completions from the ShareGPT [Chiang et al., 2023] for SFT fine-tuning, and the remaining 90% of the prompts for reward modeling and PPO fine-tuning. We use the same prompts for reward modeling and PPO fine-tuning to negate the concerns about reward overoptimization [Gao et al., 2022].

To preprocess the dataset, we use truncate every prompt to a maximum of 256 tokens, and truncate the completion such that the total length of the prompt and completion combined does not exceed 512 tokens. We use the following template for all all three stages of the PPO training.

```
\n\nHuman: {instruction}\n\nAssistant: {completion}<eos>
```

We limit the generations during preference data collection and the PPO training to a maximum of 512 tokens, including the prompt.

**SFT Fine-tuning:** We start with a pre-trained Mistral [Jiang et al., 2023] model, and fine-tune it using the single-turn instructions from the ShareGPT data [Chiang et al., 2023]. We use the same SFT checkpoints for DPO and LAIF experiments to ensure fair comparison.

**Reward Modeling:** For reward modeling, we use the same response pairs that were used for DPO training, i.e., we sample one completion in the pair from $\mathcal{M}_{\text{teacher}}$ and the other from $\pi_{\text{SFT}}$, i.e., the pair contains one completion each sampled from GPT-3.5 and from the SFT fine-tuned Mistral model. We use GPT-4 as the critic for preference data collection. We hold the last 512 preference data instances for as the validation set and use the remaining preferences for training the reward model. We trained the reward model using the Adam optimizer with the default hyper parameters, the global batch size of 8, and tried the following learning rates: {1e-4, 5e-5, 1.41e-5, 5e-6}, and trained for 5 epochs. We selected the checkpoint with the best validation accuracy. Our best reward model was trained with 5e-5 learning rate and had a validation accuracy of 84.18%. Figure 8 shows the reward modeling evaluation accuracy across steps for different learning rates.

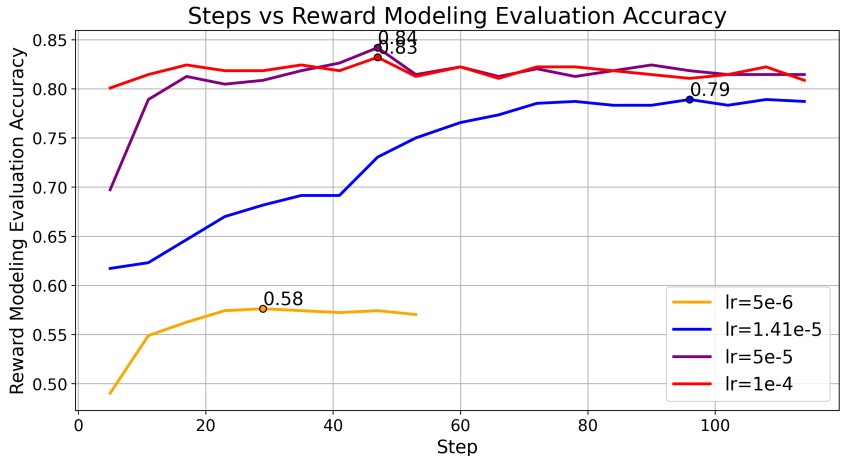

Figure 8: The highest point for each LR curve is highlighted. Our best reward model is the model trained with lr=5e-5, which reaches an accuracy of 84.18%.

**PPO Fine-Tuning:** We use the same prompts and split from the reward modeling stage for PPO fine-tuning. The prompts are truncated to a maximum length of 256, and limit the total sequence

length to 512. We use the last 512 prompts for validation. We trained our model with the Adam optimizer with the default hyper parameters, and used the adaptive KL controller from Ouyang et al. [2022] . We did the hyper parameter search on two dimensions, the initial KL coefficient (init_kl_coef), and the learning rate (lr), and used the reward model score for the checkpoint selection.

We use LoRA for PPO fine-tuning to fit our computational constraints. We used the following hyperparameters: $r = 16$, $\alpha = 32$, and a dropout of $0.05$. Figure 9 and Figure 10 show the training and validation reward score evolution during the course of the training. We find that the reward score improves over the baseline SFT model for almost all hyperparameter combinations. We select the checkpoint with the best validation reward score for the downstream AlpacaEval evaluation.

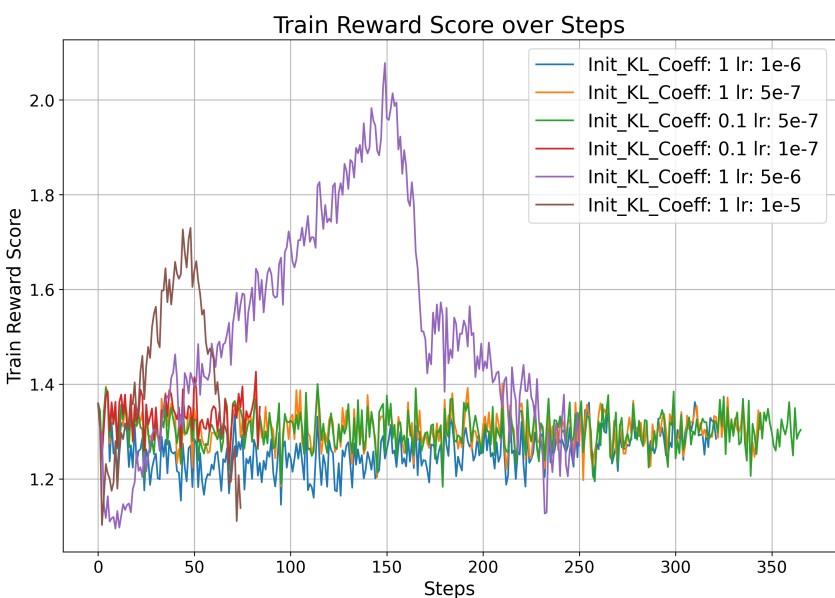

Figure 9: Train reward over PPO iterations.

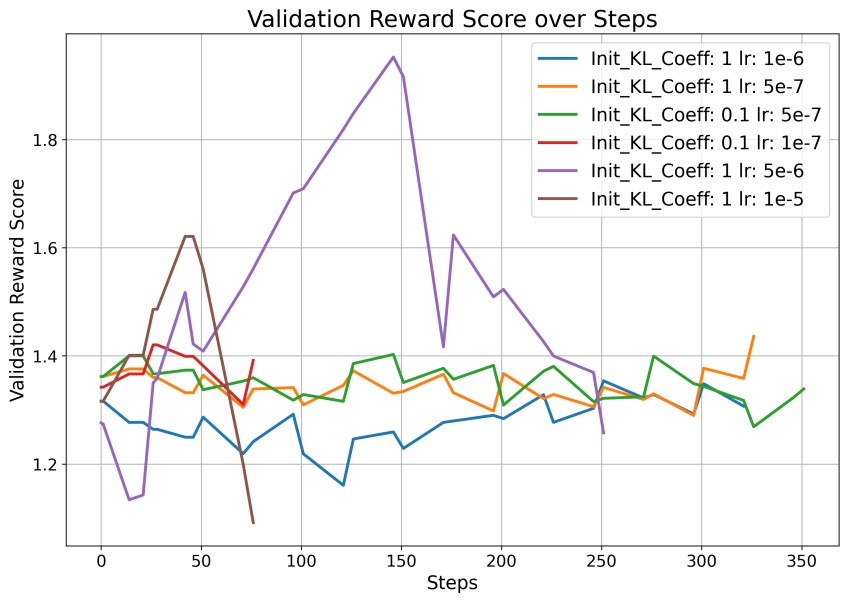

Figure 10: Validation reward over PPO iterations.

| init_kl_coef | lr | AlpacaEval Delta |
|:---:|:---:|:---:|
| 1 | 1.00E-06 | -0.9 |
| 1 | 5.00E-07 | 2.307 |
| 0.1 | 5.00E-07 | 1.767 |
| 0.1 | 1.00E-07 | 2.506 |
| 1 | 5.00E-06 | **3.361** |

Table 2: AlpacaEval score delta between the SFT and the best checkpoint according to the validation reward score for various configurations tested.

**Results**. Table 2 shows the *improvement* in the AlpacaEval win rate over the SFT checkpoint when fine-tuning using PPO. In our experiments, we observe that the PPO-AIF gives at most $3.36\%$ points improvement over the SFT model. PPO-AIF yields considerably less than the improvement (3.361 points) when compared to DPO-AIF (14.43 points), though we expect that this may be an artifact of our limited computational budget. However, this is substantially less ($\approx 20$ points) than what we achieve by just doing SFT with a data from a strong teacher such as GPT-4. Hence, the our observation that doing SFT with a stronger teacher (GPT-4) often yields better performance than doing AIF with a stronger critic (GPT-4) on a weaker SFT model (GPT-3.5 SFT), stands in the case of PPO-AIF.

