# OpenReview forum: "A Critical Evaluation of AI Feedback for Aligning Large Language Models"
_NeurIPS.cc/2024/Conference — NeurIPS 2024 poster_

### Official Review · Reviewer_6Bqz · 2024-07-02

**Soundness:** 3
**Presentation:** 4
**Contribution:** 3
**Rating:** 6
**Confidence:** 4

**Summary:**

The paper explores the current experimental set up for learning from AI feedback (LAIF), specifically using demonstrations generated by a LLM that is weaker than the LLM used to assign preference labels both for training and evaluation. By comparing training several different base LLMs on demonstrations generated by weaker (GPT3.5) versus stronger (GPT4, Claude) LLMs both in the absence of LAIF and followed by LAIF, the results suggest that current evidence for the benefits of LAIF (noted to be distinct from learning from human feedback - LHF) is over stated. The conclusion and take away is to better construct the SFT datasets such that the available demonstrations come from LLMs of the same quality level as those used to assign preference labels.

**Strengths:**

- The paper is well written and easy to follow. The arguments are well outlined to make the motivations, benefits of the work, and the learnings clear.
- The issue of ensuring that current and widely used experimental designs are correctly constructed is a vital contribution. The potential need for algorithmic adjustments to learn from AI feedback versus human feedback is something that could kick off a whole new avenue of research and investigation in the field.
- This is probably one of the few papers I have seen that attempts to dig into differences in LAIF performance gains for different base models. While the investigation is cursory, it does provide some evidence and hypotheses that can help direct future investigations, either by these authors or others.

**Weaknesses:**

- Some of the motivation for the experiment set up is incongruous. The authors motivate the paper around the LAIF paradigm (not using LLMs as human proxies for algorithm development), and call out why the claims in the paper should be considered separate from LHF. However, they also highlight the importance of having the same critic as evaluator. In practice the final evaluators for a LLM trained with LAIF methods are humans. Therefore, it would be good to have experiments where humans are the evaluators or the critic and evaluator are not the same LLM.
- A clear distinction between LAIF and LHF is made throughout the paper to highlight that the take aways are for LAIF only. However, it seems some of the take aways discussed in "Current base LLMs are insufficiently responsive to AI feedback" (pg. 7) could apply to LHF. Using strong LLMs as a proxy for humans is valid, and when the experiments are looked at from the perspective of the LAIF as proxies for LHF, more conclusions can be drawn made. Why would the hypotheses about representational space mismatches not be relevant to LHF?
- The experiment evaluating training the base LLM on demonstrations from Claude prior to LAIF is only run on llama. It would be helpful to at least also see results for Mistral 7B so that more of a comparison can be drawn against Figure 3.
- Different experiments are run with different numbers of base LLMs. The same base LLMs should be used across all main experiments, those whose results are in Figures 2, 3, 4, and 7. If fewer base LLMs are used in some experiments, a justification should be provided.
- The SFT results with GPT and Claude should not be separately compared against the LAIF results using the 10% GPT-3.5 demonstrations versus the GPT-4/Claude demonstrations.
- The story around the section "Completions samples from SFT models are substantially poorer than completions sampled from M_{oracle}" should be better motivated with a clear, strong take away.

**Questions:**

- Which version of Claude was used?
- Why not use AlpacaEval 2.0? Why evaluate against GPT-3 (text-davinci-003)?
- How do these findings connect and relate to "Preference Fine-Tuning of LLMs Should Leverage Suboptimal, On-Policy Data" (https://arxiv.org/pdf/2404.14367)?
- How did you decide which base models to use in each experiment?

**Limitations:**

The limitations are limited. There is a single sentence saying that there is only a limited analysis. However, other limitations should be discussed and addressed. For example, what are scenarios under which the analysis/results that was done may not hold? The current experimental set up assumes (implicitly) that the goal is to copy/mimic/match the performance of GPT4/Claude. However, what if someone does not want a final LLM that approximates the distribution of GPT4/Claude? What about in the base of bootstrapping versus distilling? The answers to the questions above are kind of scattered across the paper, but it would be beneficial to have a clear section discussing the boundaries of the work.

---

> ### Author Rebuttal · Authors · 2024-08-07
>
> Thanks for finding our paper well written, and our contributions for experimental design vital. We address your concerns and questions below:
>
>
> >  However, it seems some of the take aways discussed in "Current base LLMs are insufficiently responsive to AI feedback" (pg. 7) could apply to LHF. Using strong LLMs as a proxy for humans is valid, and when the experiments are looked at from the perspective of the LAIF as proxies for LHF, more conclusions can be drawn made. Why would the hypotheses about representational space mismatches not be relevant to LHF?
>
> Thanks for noting this and we agree, and it is possible that some of the base models are insufficiently responsive to HF too (and may partly explain why reward model accuracies for human feedback are low, ~70%). However, in LAIF, the discrepancy between representational strength of the teachers (like GPT-4, Claude) and models (Llama/Mistral-7B) is clearer, and we wanted to be conservative in our claims.
>
> > The story around the section "Completions samples from SFT models are substantially poorer than completions sampled from M_{oracle}" should be better motivated with a clear, strong take away.
>
> We will revise the text to include the takeaway: “Better samples from the student models, generated using better prompts or CoT and similar techniques may generate higher quality samples and allow for LAIF to be more effective”. Please let us know if rephrasing this would be better.
>
> > Which version of Claude was used?
>
> We used Claude-v1. Claude-v2 was found to have a poorer correlation with human judgment in the original AlpacaEval study.
>
> > How do these findings connect and relate to "Preference Fine-Tuning of LLMs Should Leverage Suboptimal, On-Policy Data" (https://arxiv.org/pdf/2404.14367)?
>
> The findings are complementary, as the main message of the cited paper is how to sample data for labeling or query reward model in RLHF, whereas our paper analyzes how effective automatic mechanisms such as AI feedback (loosely, our paper studies the choice of reward model and the paper cited in the question studies the data used during optimization).
>
> > How did you decide which base models to use in each experiment?
>
> Our academic access to Anthropic API expired and was not renewed, so our studies with Claude are more limited as compared to GPT-4.

---

> > ### Comment · Reviewer_6Bqz · 2024-08-08
> >
> > Thank you for addressing my question and concerns. I will raise my score accordingly

---

### Official Review · Reviewer_pKmA · 2024-07-13

**Soundness:** 3
**Presentation:** 3
**Contribution:** 3
**Rating:** 6
**Confidence:** 3

**Summary:**

The paper evaluates the extent to which AI feedback is helpful in aligning large language models (LLMs) within the commonly used two-step method of improving pre-trained LLMs. This method involves first performing supervised fine-tuning (SFT) and then fine-tuning with reinforcement learning (RL) or direct preference optimization (DPO) using preference feedback. The findings indicate that, in some cases, SFT may outperform the two-step LAIF approach.

**Strengths:**

- The paper includes comprehensive experiments and provides a detailed analysis along with hypotheses explaining the results.
- These experiments encompass a wide range of different LLMs and settings, offering good quantitative metrics and analyses.
- I particularly appreciate the bandit experiments; despite their simplicity, they effectively convey the core ideas and strongly support the paper's claims.

**Weaknesses:**

- I am not entirely certain about the claim that “SFT on strong distribution minimizes any improvements from LAIF.” While this was the case for 3 out of the 4 result settings (including both figures 4 and 7), it is difficult to assert this as a general truth. Could the authors rephrase the claim to be more nuanced?
- In the analysis, the authors make general claims that may not hold true in all cases. For example, “AI feedback is effective when there is a discrepancy between the SFT distribution and the evaluator,”, the analysis lacks numerical values, and the claims are not nuanced, even though they do not hold in all cases. However, I do like the conclusion, where the authors emphasize that the claim does not hold true in all cases.
- It would be interesting to see if the same hypotheses hold in more general LLM settings, such as multi-turn instructions and multi-modal foundation models.

Minor things
- Typo in line 331 “via via LAIF”

**Questions:**

- In Table 1, are the values in brackets variances or confidence intervals? How many repeated runs were conducted?
- When the paper uses phrases like “weaker SFT target distribution,” how exactly is a target distribution determined to be weaker or stronger?
- It seems that the difference in target distribution is based solely on the percentage of total examples used. It would be interesting to see if the diversity of examples affects the improvements from LAIF and to what extent. For instance, what if the total number of examples remains the same, but the diversity of examples in terms of their distribution in an embedding space is different?

**Limitations:**

- Yes, the authors have sufficiently addressed the limitations.

---

> ### Author Rebuttal · Authors · 2024-08-07
>
> Thanks for finding our experiments comprehensive and appreciating our bandit experiments! We answer your questions and concerns below:
>
> > I am not entirely certain about the claim that “SFT on strong distribution minimizes any improvements from LAIF.” While this was the case for 3 out of the 4 result settings (including both figures 4 and 7), it is difficult to assert this as a general truth. Could the authors rephrase the claim to be more nuanced?
>
> > In the analysis, the authors make general claims that may not hold true in all cases …
>
> Thanks for the suggestions. We will revise the text to be nuanced, and point the readers to conclusions explicitly for takeaways. We will revise the specific statement to be more nuanced: “We found LAIF to provide minimal improvement when SFT was a dataset of completions from a strong teacher, in 3 out of the 4 cases”.
>
> > In Table 1, are the values in brackets variances or confidence intervals? How many repeated runs were conducted?
>
> The confidence intervals were computed by the automated evaluation in AlpacaEval, which evaluates models on 800 questions (we will clarify this in our text). Our computational budget did not allow for repeated runs per model/dataset/teacher, and we favored distributing our budget over more models and teachers than repeated runs with a smaller set of models and teachers.
>
> > When the paper uses phrases like “weaker SFT target distribution,” how exactly is a target distribution determined to be weaker or stronger?
>
> Thanks for bringing this up, and we recognize the impreciseness of the notion of strength in this context. The strength of the target distribution in this context is used to refer to the strength of the teacher model, and GPT-3.5 is generally agreed to be worse at instruction following compared to GPT-4 (both subjectively, but also benchmarks like LMSys and AlpacaEval).
>
> > It would be interesting to see if the diversity of examples affects the improvements from LAIF and to what extent. For instance, what if the total number of examples remains the same, but the diversity of examples in terms of their distribution in an embedding space is different?
>
> That’s a great suggestion, and we agree it would be good to explore in future work!

---

> > ### Comment · Reviewer_pKmA · 2024-08-08
> >
> > Thank you for addressing my concerns and questions! I will keep my current score.

---

### Official Review · Reviewer_Sb9E · 2024-07-14

**Soundness:** 3
**Presentation:** 3
**Contribution:** 3
**Rating:** 6
**Confidence:** 3

**Summary:**

Learning from AI Feedback (LAIF) has become a popular alternative for improving the instruction-following abilities of large language models (LLMs). Despite its popularity, many unresolved questions remain regarding the actual improvements gained through LAIF. The authors address some of these questions, with a focus on where exactly the improvement in the LAIF pipeline is coming from

The authors found that many of the improvements in LAIF are attributed to the differences in the weak teacher LLM (that provides the SFT data) and a strong critic LLM (that provides the preference data). Their empirical evidence demonstrated this issue across a wide range of base, critic, evaluator, and teacher models.

Moreover, the authors suggested two potential explanations for LAIF's ineffectiveness: either the preference dataset is not sufficiently informative, or the base model has inherent limitations. Finally, the authors offer several insightful suggestions for future research in the area of LAIF.

**Strengths:**

- The paper is well-written and easy to follow.
- The authors are addressing a significant problem.
- The experiments were really well designed and performed.
- The authors show the robustness of their observation by running experiments across several models and dataset splits.
- The authors not only identified the problem in LAIF but also provided some possible explanations that enhanced the reader's understanding of it.
- LAIF is an important path forward for improving LLM instruction following capability. Therefore, as outlined in the paper, it is important to systematically identify the problems in LAIF so that researchers can address them.
- The authors address most of my obvious internal questions and thoughts on various experiment details and design decisions.

**Weaknesses:**

- Some of the observations in the paper are straightforward.
- A few more experiments should be included in the paper to complete some of its conclusions.
- The 10% rule doesn't always hold true. In Figure 3 and Figure 4, SFT 10% performs worse than SFT 100%. A better split could improve SFT performance when doing SFT + LAIF, which is important based on the paper's conclusions.
- The LAIF ablation experiments for addressing the LAIF ineffectiveness, either as preference data or as the base model, have issues. The author samples data from Llama or Mistral and trains on the data using Llama or Mistral as the baseline model. However, the results could suffer from overoptimization problems [1].

[1] Scaling Laws for Reward Model Overoptimization by Gao et al 2022

**Questions:**

- Line 64: If the discrepancy between SFT and AI feedback is minimal, then doing SFT can suffice. I am trying to understand what this statement implies; it seems pretty straightforward. If there is no gap between the two, then you do not need the second step.
- Line 206: Missing citation, using completions from M_{teacher} as one of the inputs in the preference pair results, was observed in [1].
- Also, why not use the M_{teacher} to generate both responses to ensure that the preference data is high quality? DPO [2] mentioned this setting in the "DPO outline" section; essentially, you can train on \pi_{ref} on the preferred completions.
- Line 252: Would you agree that M_{critic} does not fully capture the quality of the preference dataset? If so, then comparing M_{critic} versus M_{teacher} is a little odd because M_{teacher} fully affects the quality of the SFT dataset, whereas M_{critic} only partially affects the quality of the dataset. If M_{critic} generated both the chosen and rejected, then you be certain that generations are high quality and the M_{critic} learning signal is good.
- Figures 4 and 7 show that the 10% threshold is ideal for certain models and settings. Have you experimented with a different percentage threshold?
- Missing citations [3], [4], [5]

[1] Coactive Learning for Large Language Models using Implicit User Feedback by Tucker et al. ICML 2024

[2] Direct Preference Optimization: Your Language Model is Secretly a Reward Model by Rafailov et al. NeurIPS 2023

[3] Starling-7B: Increasing LLM Helpfulness & Harmlessness with RLAIF by Zhu et al. 2023

[4] UltraFeedback: Boosting Language Models with High-quality Feedback by Cui et al. 2023

[5] Peering Through Preferences: Unraveling Feedback Aquisitiosn for aligning Large Language models by Bansal et a. 2024 ICLR

**Limitations:**

Yes, the authors have adequately addressed the limitations.

---

> ### Author Rebuttal · Authors · 2024-08-07
>
> > 10% rule doesn’t always hold
>
> > Figures 4 and 7 show that the 10% threshold is ideal for certain models and settings. Have you experimented with a different percentage threshold?
>
> Thanks for noting this. Using an increasingly larger split for SFT would weaken our claims compare LAIF with SFT (for example, consider using 99% of the prompts for SFT and 1% for LAIF). We heuristically chose a split which provides enough SFT data to provide a good initialization for LAIF, while using most of the prompts for LAIF. We will revise the text to note that this heuristic may not be optimal.
>
> > Line 64: If the discrepancy between SFT and AI feedback is minimal, then doing SFT can suffice. I am trying to understand what this statement implies; it seems pretty straightforward. If there is no gap between the two, then you do not need the second step.
>
> That is the right interpretation → if you do not see any substantial improvement from LAIF, it was not needed in the first place. We will rephrase this sentence to be clearer.
>
> > Line 206: Missing citation, using completions from M_{teacher} as one of the inputs in the preference pair results, was observed in [1].
>
> > Missing citations [3], [4], [5]
>
> Thanks for the pointers. We will add the missing citations in the next revision.
>
> > Also, why not use the M_{teacher} to generate both responses to ensure that the preference data is high quality? DPO [2] mentioned this setting in the "DPO outline" section; essentially, you can train on \pi_{ref} on the preferred completions.
>
> This is a great suggestion, and would be good to explore better in future work. There are two concerns:
> - If generating a completion for SFT from the teacher costs 1 unit of supervision per prompt, and generating a preference label (given two completions) costs 1 unit of supervision, our current setup for LAIF uses 2 units of supervision per prompt, compared to SFT which only gets 1 unit per prompt. Generating both the completions using the teacher would mean 3 units of supervision per prompt. We ideally want to compare SFT and LAIF with equivalent amounts of supervision, but our setup is already unfairly biased to LAIF.
> - Our preliminary experiments find that sampling both the completions from the teacher underperforms using our current scheme 1 completion from the teacher and 1 from the model. DPO [1] discusses how samples not from the reference model can hurt the performance. There may be ways to improve the pipeline when both the samples are from the teacher (for example, training pi_{ref} on both preferred and dispreferred completions), it requires deeper exploration beyond the scope of this paper.
>
> [1] Direct Preference Optimization: Your Language Model is Secretly a Reward Model by Rafailov et al. NeurIPS 2023

---

> > ### Comment · Reviewer_Sb9E · 2024-08-12
> >
> > Thank you for addressing my concerns and questions! I will keep my current score.

---

### Decision · Program_Chairs · 2024-09-25

**Decision:**

Accept (poster)

**Comment:**

The paper carefully studies the current practice of Learning from AI Feedback (LAIF) for fine-tuning large language models and show that the benefits of LAIF (distinct from learning from human feedback) are likely over-stated. All of the reviewers found the experiments comprehensive and convincing, and the paper's highlighting of the need for better Supervised Fine-Tuning datasets (should be drawn from LLMs of the same quality as the LLMs used to assign preferences) is likely to be of value to the LLM fine-tuning and distillation community.

The reviewers raised several questions that led to insightful discussions with the authors. All of the reviewers' questions were eventually answered satisfactorily, and incorporating all of the discussions into a revised version will substantially strengthen the paper.